# Study of Concussion in Rugby Union through MicroRNAs (SCRUM): a study protocol of a prospective, observational cohort study

Kamal M Yakoub,[1,2] Patrick O'Halloran,[2] David J Davies,[1,2] Conor Bentley,[1] Callum N Watson,[2] Mario Forcione,[1,2] Ugo Scarpa,[2] Jonathan R B Bishop,[1] Emma Toman,[3] Douglas Hammond,[3] Matthew J Cross,[4,5] Keith A Stokes,[4,6] Simon P T Kemp,[6] David K Menon,[7] Valentina Di Pietro,[1,2,8] Antonio Belli[1,2]

For numbered affiliations see end of article.

**Correspondence to**
Dr Kamal M Yakoub;
k.yakoub@bham.ac.uk

## ABSTRACT

**Introduction** The diagnosis of mild traumatic brain injury or sports-related concussion is a challenge for all clinicians, players, coaches and parents involved in contact sports. Currently, there is no validated objective biomarker available to assess the presence or severity of concussion in sport, and so it is necessary to rely on subjective measures like self-reporting of symptoms which depend on the cooperation of the athlete. There is a significant health risk associated with repetitive injury if the diagnosis is missed, and so there is great value in an objective biomarker to assist diagnostic and prognostic decisions.

**Objective** To establish a panel of non-invasive MicroRNA biomarkers in urine and saliva for the rapid diagnosis of sports-related concussion and investigate the kinetics and clinical utility of these biomarkers in assisting diagnostic, prognostic and return-to-play decisions.

**Methods and analysis** Observational, prospective, multicentre cohort study recruiting between the 2017–2018 and 2018–2019 Rugby Union seasons. Professional rugby players in the two highest tiers of senior professional domestic rugby competition in England will be recruited prospectively to the study. During the season, three groups will be identified: athletes entering the World Rugby Head Injury Assessment (HIA) protocol, uninjured control athletes and control athletes with musculoskeletal injuries. Saliva and urine will be collected from these athletes at multiple timepoints, coinciding with key times in the HIA protocol and return-to-play process.

**Ethics and dissemination** Ethics approval has been obtained. The compiled and analysed results will be presented at national and international conferences concerning the care of patients with traumatic brain injury. Results will also be submitted for peer review and publication in the subject journals/literature.

## INTRODUCTION

Sports-related concussion is a form of mild traumatic brain injury caused by biomechanical forces transmitted directly or indirectly to the head and neck. Data from the Rugby Football Union (RFU) indicate that concussion occurs at an incidence of 20.9/1000 player

## Strengths and limitations of this study

► This study will contribute to a better understanding of the pathophysiology of sports-related concussion and the role of MicroRNAs within this.
► The study will prospectively recruit elite Rugby Union athletes entering the World Rugby Head Injury Assessment (HIA) protocol, before their diagnosis of sports-related concussion is confirmed, as well as elite athlete controls both with and without musculoskeletal trauma injuries.
► This work will potentially provide a novel means of non-invasively and objectively diagnosing sports-related concussion and advising on decisions relating to symptomatic treatment and return-to-play.
► The study will allow for the modelling of this novel diagnostic tool in relation to an existing, robust and widely used multimodal concussion assessment process (the World Rugby HIA protocol) to understand the value which might be added by salivary and urinary markers.
► This study will recruit from a population of elite male adult athletes and so applications to other populations, such as female and youth athletes, will require further study.

hours in elite-level Rugby Union in England.[1] In the majority of cases, concussion results in a rapid onset, but short-lived, impairment of neurological function that resolves spontaneously.[2] It is accompanied by an absence of findings on conventional neuroimaging and, while 80%–90% of the sufferers are able to return to preinjury levels of neurocognitive and physical functions within 2 weeks, 2.5% go on to experience persistent postconcussion symptoms (eg, headaches, dizziness, fatigue, memory problems).[3 4]

Concussion incidence has increased in Rugby Union over recent years,[1] however it is felt that this, at least in part, reflects a rising

awareness of the problem, rather than a true increase in rate. This mirrors a significant interest in this injury in the popular press generated in part by stories of catastrophic short-term consequences and persistent disability following concussions or repeated concussions along with extensive litigation in the USA.[2 3 5]

Despite the popular media attention surrounding concussion, there remain a number of unanswered questions with regard to both the short-term and long-term consequences of this condition.[6] The relationship between the cumulative burden of concussive and sub-concussive injuries and significant clinical outcomes remains uncertain. These outcomes include the later development of neurodegenerative disease such as chronic traumatic encephalopathy, the potential for catastrophic injury (second impact syndrome) following a premature return to head contact activities following concussion and the need for rehabilitation of patient suffering prolonged post concussive symptoms; all of which require further exploration. The issue of safe return to sport following concussion is of particular note here in light of evidence of a prolonged period of metabolic disturbance[7] and increased risk of subsequent injuries when returning to play,[8] both of which outlast symptomatic recovery.

In part, the difficulty in answering these questions fully relates to the challenge of diagnosing concussion itself which continues to be an issue across all sports. It is recognised that the on-field assessment of players for signs and symptoms of concussion is challenging,[9 10] and that there is at present no perfect test with which to diagnose concussion.

In order to combat current diagnostic uncertainty, expert consensus recommends the use of multimodal diagnostic tools, at multiple timepoints to allow for the varied and evolving presentations of concussion to be picked up by clinicians.[9–13] However, the need for player engagement, the availability of clinicians, the need for baseline athlete information and the variation in diagnostic threshold between clinicians means that it is still challenging to establish a definitive diagnosis of sports-related concussion in athletes.

One example of sport attempting to facilitate diagnosis of concussion is the Head Injury Assessment (HIA) protocol in elite-level Rugby Union. This protocol has been described previously and has been evaluated and shown to reduce the proportion of players being allowed to continue playing after suffering a concussion.[11 12]

Briefly, players who demonstrate clear observable signs or symptoms of concussion (defined as conforming to any one of 11 'criteria 1' signs or symptoms) following a head injury event either observed directly by medical staff or following video review or confirmed during an on pitch medical assessment (eg, loss of consciousness, tonic posturing, nystagmus) are permanently removed from the game without any further off-field concussion screening and, irrespective of further clinical evaluation, are considered to have sustained a concussion.

Where the outcome of a head impact event is not clear, for example if no criteria 1 signs or symptoms are identified but there is a suspicion that the injury mechanism may have resulted in a concussion (defined as conforming to a set of 'criteria 2' signs), the players undergo a standardised off-field assessment. This off-field screening assessment takes the form of the multimodal 'HIA1' assessment involving Maddock's questions, symptom checklist, immediate and delayed recall, tandem gait test and assessment of clinical signs. A window of 10 min is permitted for this assessment to be conducted with the player being temporarily replaced in the game for that period. Any abnormality in the HIA1 assessment and/or a clinical suspicion of concussion by the assessing practitioner mandates a permanent removal from play, but a diagnosis of concussion is not made or excluded at this time point. An electronic decision aid is used to assist this assessment with reference to baseline data collected in preseason for each player.

All players who undergo the HIA1 assessment, irrespective of outcome, have more detailed clinical assessments postmatch (HIA2) and again after two nights rest (HIA3) to monitor their progress and confirm or exclude a diagnosis of concussion. The HIA2 assessment involves the use of the Sport Concussion Assessment Tool 5th edition,[14] tandem gait test and clinical assessment, while the HIA3 may also include the use of neurocognitive testing (typically computerised tools such as Cogsport).

The diagnosis of concussion then is reached either during the game on the basis of the presence of standardised criteria 1 features, or in the 48 hours following the match on the basis of two subsequent clinical assessments by the team physician at the HIA2 and HIA3 timepoints, supported by the standardised HIA2 or HIA3 assessments conducted by the team physician.

Therefore this process divides players into three groups:
► Players who are immediately and permanently removed from the game on the basis of criteria 1 features (immediate permanent removal) and diagnosed with concussion.
► Players who enter the HIA1 screening process and are not diagnosed with concussion following the completion of either of their HIA2 and HIA3 assessments.
► Players who enter the HIA1 screening process and are diagnosed with concussion at either their HIA2 or HIA3 assessments.

Away from elite sport, there is, if anything, more variation in the diagnostic thresholds and safe return-to-activity advice which is given to patients and military personnel. This is reflected in the most recent UK National Institute for Health and Care Excellence guidance which has highlighted the importance of traumatic brain injury biomarker research in head injury.[15]

MicroRNAs (miRNA) are a recently discovered class of non-coding RNAs that play key roles in the regulation of gene expression. They are attracting increasing interest in clinical research as potential biomarkers for the identification and classification of cancers as well as other disease states including neurodegenerative diseases and most recently, brain trauma.[16–18] This is reflected in the

exponentially increasing number of publications on their use in a variety of medical applications.

Work by DiPietro and colleagues has demonstrated the value of serum miRNA in providing prognostic information regarding the severity of traumatic brain injury in road traffic collision patients. Furthermore, this has been shown to be of value from very early timepoints, including at the roadside.[16] In light of the potential challenges of using an invasive serum test in an athletic setting, follow-up work in the Birmingham Sport Concussion Clinic at the Queen Elizabeth Hospital Birmingham (QEHB) has looked at the presence of miRNA markers in the blood, saliva and urine of concussed athletic patients presenting to the clinic within 72 hours of injury and healthy controls. This has identified five miRNA markers which are upregulated in saliva following concussion, demonstrating the potential promise of this category of fluid biomarkers in assisting with this diagnostic challenge.[19] Therefore, our objective is to establish a panel of non-invasive miRNA biomarkers in urine and saliva for the rapid diagnosis of sports-related concussion and investigate the kinetics and clinical utility of these biomarkers in assisting diagnostic, prognostic and return-to-play decisions.

## METHODS AND ANALYSIS
### Study design
The Study of Concussion in Rugby Union through MicroRNAs (SCRUM) is a prospective observational cohort study of all male players (up to 1100) in the two highest tiers of senior professional domestic rugby competition in England, the Premiership and Championship. The study is embedded within the ongoing RFU's Professional Rugby Injury Surveillance Programme for the 2017–2018 and 2018–2019 seasons.

All players will be visited at their club during the preseason period and provided with information about the study, including the processing and handling of their samples, the anonymisation of their data and their right to withdraw their voluntary consent at any time, before being asked to complete a consent form. They will then be asked to provide a single baseline sample of approximately 2 mL of saliva and about 5 mL of urine, if possible before their usual training session. These samples will provide the baseline miRNA values for a large population of elite athletes and are taken alongside a baseline questionnaire which provides information on previous concussion history and symptoms.

In addition to this baseline collection from the players, the club medical teams will be visited and provided with an explanation of the study materials and sample-collection procedures. This will be reinforced with regular email reminders during the season, the availability of the study team via telephone or email 24/7 to answer questions and an online video demonstrating the collection procedure (see online supplementary video).

During the season, each time a player enters the HIA process he will be asked for saliva sample at the HIA1 timepoint and for saliva and urine samples at the HIA2 and HIA3 timepoints. In this way, we will be able to investigate the presence of concussion diagnostic miRNAs non-invasively at multiple timepoints which are matched to a standardised clinical assessment and diagnostic threshold for concussion performed by a clinician with experience in rugby head injuries.[12]

Players diagnosed with concussion will be asked for a sample of saliva and urine at the point at which they are cleared by their team physician to return to rugby (stage V) as per the Berlin Consensus Graduated Return-to-Play protocol.[20] This will allow for the time course for recovery of the identified miRNA markers to be mapped. Data regarding the identification of players entering the HIA process and returning to play are collected weekly by the RFU and Premiership Rugby which will allow us to monitor compliance through the season and to analyse differences between incidents included and not included in our data collection.

Whenever a player enters the HIA process, club medical teams will also be asked to approach an 'uninjured control' player for saliva and urine samples at the HIA2 and HIA3 timepoints. An 'uninjured control' is a non-concussed player who has played approximately a matched number of minutes in the same game. This will allow for the description of miRNA profiles in response to the physical stress of athletic activity and exposure to the likely sub-concussive blows which come from participation in a collision sport.

To rule out contamination from non-neurological injuries and musculoskeletal trauma, club medics will also be asked to collect samples from players removed from the field of play due to musculoskeletal injuries (orthopaedic controls) at the same timepoints as the 'uninjured control' group. The matching ratio for these groups (concussion, uninjured controls and orthopaedic controls) will be 1:1:1 (see table 1).

The samples will be collected in Oragene-RNA saliva collection pots and Norgen Biotech urine sample pots, both of which contain a proprietary miRNA stabilising solution. The samples will be transported to the University of Birmingham (UoB) where they will be processed in line with the sample manufacturer's guidance to allow freezing and storage.

A next-generation sequencing (NGS) procedure will be carried out by the RNA sequencing services of Exiqon on an unbiased preliminary selection of an estimated 20 samples in each of the different players groups. NGS allows the identification of all potential non-coding RNAs that are differentially expressed between study groups. The NGS analysis will potentially identify a number of suitable candidate biomarkers. To correct for the effect of multiple comparisons, we will apply the Benjamini-Hochberg procedure to select biomarkers with a false discovery rate <0.05 for further analysis by quantitative PCR in the whole population of study samples. From the surviving panel, we will exclude biomarkers whose concentrations are not significantly different from matched baseline values ($p > 0.05$).

**Table 1** SCRUM sample collection schema

| Test | Baseline | Pitch-side | Postmatch | 36–48 hours postinjury | Stage V GRTP |
|------|----------|-----------|-----------|------------------------|--------------|
| **Concussed player** | | | | | |
| HIA | X | X (HIA1) | X (HIA2) | X (HIA3) | |
| Saliva | X | X | X | X | X |
| Urine | X | | X | X | X |
| **Orthopaedic control player** | | | | | |
| HIA | X | | | | |
| Saliva | X | | X | X | |
| Urine | X | | X | X | |
| **Uninjured player** | | | | | |
| HIA | X | | | | |
| Saliva | X | | X | X | |
| Urine | X | | X | X | |

GRTP, Graduated Return-to-Play protocol; HIA, Head Injury Assessment protocol; SCRUM, Study of Concussion in Rugby Union through MicroRNAs.

### Sample size

The RFU estimates that out of the 1100 players participating in the top two tiers of English rugby (The Premiership and Championship) each season, 350–400 players enter the HIA process, of whom approximately 200 will be diagnosed as concussed. If participation was universal, the study would generate approximately 3500 saliva samples and 2900 urine samples from the following cohorts during the course of a season:

► 1100 players tested preseason.
► Approximately 200 players will provide repeat baseline samples in the postseason. This will allow for quality control for intraindividual variation from season to season.
► 200 players with a definitive diagnosis of concussion following either identification of criteria 1 signs in-game or following the HIA2 and HIA3 assessments in the 48 hours postgame.
► 150–200 players temporarily removed from play for an off-field assessment following a head injury event (possible concussion) but for whom concussion is not subsequently confirmed following the HIA2 and HIA3 assessments in the 48 hours postgame.
► 200 matched controls with musculoskeletal injuries tested twice.
► 200 matched uninjured controls tested twice.

### Outcome measures and statistical analysis

Biomarker data will be analysed with standard group comparison statistics (eg, t-test on normalised data or Mann-Whitney on non-normalisable data).

### Primary outcome(s)

The primary outcome measure will be the postmatch (HIA2) comparison of levels of a panel of miRNAs in saliva and urine samples from players who have been definitively diagnosed with concussion through the HIA process described above versus players in the uninjured and musculoskeletal injury control groups described above.

### Secondary outcomes

Secondary outcomes are:
► Kinetics of the sample biomarkers at the different timepoints outlined in the protocol above in table 1.
► We will also report HIA1, 2 and 3 levels of the same miRNA panel for players entering the HIA process who were not eventually diagnosed with concussion.
► Correlation between levels of specific biomarkers and severity, type and duration of symptoms as well as time to return to full participation.

### Patient and public involvement

The Rugby Players' Association is the representative body for professional players in England and has been involved from the outset, with members asked to share their views on the design and setting up of this study. They are also part of the steering and oversight group. In terms of dissemination, we will keep clubs updated on the progress of the study and will report the findings by email/newsletter and in various meetings where club representatives are in attendance.

### ETHICS AND DISSEMINATION

All data will be collected and kept on National Health Service servers in accordance with the 1998 UK Data Protection Act, UoB and QEHB data handling and maintenance guidelines, with the minimum amount of required information recorded. The computers on this

network have restricted physical access; data are stored under coded file names and the local network has secure password access restricted to a limited set of people.

All the compiled and analysed results will be presented at national and international conferences concerning the care of the patients with traumatic brain injury. Results will also be submitted for peer review and publication in the subject journals/literature.

The datasets generated during and/or analysed during the current study are/will be available on request from AB who is the point of contact and will be able to provide anonymised data with sufficient details to reproduce the analyses for up to 10 years.

**Author affiliations**
¹NIHR Surgical Reconstruction and Microbiology Research Centre, University Hospitals Birmingham NHS Foundation Trust, Birmingham, UK
²Neurotrauma and Ophthalmology Research Group, Institute of Inflammation and Aging, University of Birmingham, Birmingham, UK
³Head Injury Management Research Group, Faculty of Clinical and Biomedical Science, School of Dentistry, University of Central Lancashire, Preston, UK
⁴Department for Health, University of Bath, Bath, UK
⁵Premiership Rugby, Twickenham, London, UK
⁶Rugby Football Union, Twickenham, London, UK
⁷Division of Anaesthesia, Addenbrooke's Hospital, University of Cambridge, Cambridge, UK
⁸The Beckman Institute for Advanced Science and Technology, University of Illinois at Urbana Champaign, Illinois, USA

**Acknowledgements** The authors acknowledge UoB and NIHR SRMRC for their support. Also, they would like to thank the RFU, RPA and all the rugby clubs that agreed to participate in the study for their valuable feedback on the design and setting up of the study.

**Contributors** AB and VDP are the principal investigators. AB, VDP, SPTK, MJC and JRBB developed the protocol. AB, JRBB, DH, MJC, SPTK, DKM, VDP, KAS and DJD drafted the protocol and commented on the protocol. KMY, POH, DJD, CB, MF, CNW, US, ET and VDP drafted and finalised the manuscript. All authors have approved the final manuscript.

**Funding** This work is supported by funding received from the Medical Research Council (MRC), National Institute for Health Research (NIHR), Biovici Diagnostics Ltd and the Rugby Football Union (RFU), grant number: MRC CIC 164132.

**Competing interests** The University of Birmingham owns intellectual property for some of the biomarkers used in this study.

**Patient consent** Not required.

**Ethics approval** The SCRUM study runs under University of Birmingham (UoB) ethics (ERN_11-0429AP28) as part of the Repetitive Concussion in Sport (RECOS) study. This study was also approved by the East of England - Essex Research Ethics Committee (REC) on 22 September 2017 – REC 17/EE/0275; IRAS 216703. This study will also use data collected by the RFU for injury surveillance, our use of which has been approved by University of Bath Research Ethics Approval Committee for Health (EP 16/17 286).

**Provenance and peer review** Not commissioned; externally peer reviewed.

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
