## [Reviewer comments · BMJ Open]

ARTICLE DETAILS

TITLE (PROVISIONAL)	Study of Concussion in Rugby Union through MicroRNAs (SCRUM): A study protocol of a prospective, observational cohort study.
AUTHORS	Yakoub, Kamal; O'Halloran, Patrick; Davies, David; Bentley, Conor; Watson, Callum; Forcione, Mario; Scarpa, Ugo; Bishop, Jon; Toman, Emma; Hammond, Douglas; Cross, MJ; Stokes, Keith; Kemp, Simon; Menon, David; Di Pietro, V; belli, antonio

VERSION 1 – REVIEW

REVIEWER	Yushan Wang DRDC Suffield Research Centre Medicine Hat, Alberta Canada
REVIEW RETURNED	05-Jun-2018

GENERAL COMMENTS	This study protocol addresses important questions in diagnosing concussion. Overall this protocol is written in a very organized way with clear objectives. I believe this protocol should be accepted for publication after correcting a few typos and addressing the minor concerns. Minor comments:  1) Pg 6, Ln 18: Complied? compiled? 2) Page 8, Ln 6: ... is "a" form of 3) I found it odd that all references are cited after a period or a comma, is it a requirement for this journal? 4) Pg8, Ln 18-22: These two sentences need a little more clarification. For example, what are post concussion symptoms? migaine headache? cognitive functions such as learning and memory problems? 80-90% of sufferers are able to return to pre-injury levels in terms of what? symptoms? cognitive functions? or physical activities?
---

REVIEWER	Paolo Boffano Aosta hospital
REVIEW RETURNED	07-Jun-2018

GENERAL COMMENTS	Interesting and well written.
-------------------------------

REVIEWER	Dr Sarah Mc Fie University of Cape Town, Cape Town, South Africa.
REVIEW RETURNED	20-Jun-2018

GENERAL COMMENTS	The study is addressing an important area of research and has an ideal platform and methodology to succeed in the investigation. There are several minor revisions that I would advise to improve the overall quality and clarity of the paper. Abstract:
--

In the methods and analysis section, the three groups of players (uninjured, orthopaedic and concussed) could be described more clearly.

Article summary

For the last point of the article summary, I would also include younger athletes as a population that requires further study, as this is an important group that displays differences in concussion symptom presentations. So I would suggest saying "such as female or youth athletes".

Introduction

In the first sentence of the introduction, "sport related concussion" is capitalised, but in the abstract it is in lower caps and then abbreviated. This needs to be kept consistent.

The first sentence reads: "Sport Related Concussion is form of mild traumatic brain injury caused by..", this should be changed to "Sport Related Concussion is a form of mild traumatic brain injury caused by..".

In the second sentence there is a space between "England" and the punctuation.

The third sentence needs to be grammatically revised. Possibly change to: "In the majority of cases, concussion results in a rapid onset, but short-lived, impairment of neurological function that resolves spontaneously."

The fourth sentence is somewhat contradictory, as it presents two conflicting pieces of information. It is stated that 80-90% of concussed athletes return to pre-injury levels within two weeks, while 20-30% experience prolonged symptoms. both statements can't be simultaneously true.

The fourth sentence should also read: "pre-injury levels within two weeks"

The second paragraph of the introduction is one long sentence. This could be split into two or three sentences.

In paragraph five, "baseline information for athletes" can be changed to "baseline athlete information".

In the first sentence of paragraph 6, the word "process" might not need to be capitalised.

For the description of the HIA process in paragraph six, I would explain where it is currently being used after the first sentence and before going into a description of it. I feel that one needs the context of it being used in professional leagues to understand the description of video replays etc.

In paragraph nine, I would add more biological information on why miRNA's are being suggested as biomarkers. For example, highlighting the temporal expression of miRNA and why/how quantifying these levels may provide an indication of the extent of

underlying gene expression and the biological processes initiated by head injury. This may give the reader a better understanding of why miRNA analysis may be a better biomarker candidate in comparison to other gene or protein expression assays.

It might also be worth explaining how miRNA's are distributed in bodily fluids, how they interact with the blood brain barrier and the relevance of these dynamics to concussion.

In general, I think more detail needs to be provided about miRNA's and how they function to give readers without a knowledge of genetics or molecular biology a better understanding of why they are potential biomarkers.

Methods and analysis

In the first sentence of the methods the number of participants is written as "1100", while in the sample size it is written as "1,100".

In the first sentence, I would also add that the two leagues consist of senior professional players. Some international readers may not be familiar with the Premiership and Championship. For example, maybe say "the two highest tiers of senior professional domestic rugby competition", or something similar.

Will the baseline miRNA samples be taken at a certain time of day to control for circadian fluctuations? Or possibly at multiple time points to provide an "average" baseline level? See Heegaard et al. 2016: "Diurnal Variations of Human Circulating Cell-Free Micro-RNA".

It will be important to record the timing of these baseline extractions in relation to preseason training sessions, other injuries, or supplement/medication use. There are so many potential confounding factors that getting a true "baseline" from a single extraction will always be difficult.

In the fourth paragraph, "in match HIA" could be changed to "pitch-side HIA" and "36-48 HIA" could be changed to "36h-48h HIA", to be consistent with Table 1.

Paragraphs five and six could be joined into a single paragraph. Both paragraphs consist of a single, slightly long-winded, sentence.

In the first sentence of paragraph seven, shown below, it is not especially clear which athletes are being referred to.

"In the case of players who are diagnosed with concussion at any time point, they will also be asked for a sample of saliva and urine at the point at which they are cleared by their team physician to return to rugby (Stage V) as per the Berlin Consensus Graduated Return-to-Play (GRTP) protocol.[2]"

The sentence could be made more succinct, for example: "Players diagnosed with concussion will be asked for a sample of saliva and urine at the point at which they are cleared by their team physician to return to rugby (Stage V) as per the Berlin Consensus Graduated Return-to-Play (GRTP) protocol.[2]"

Will the playing position of the uninjured control be taken into account? For example, using a scrum-half as a comparison for a

	forward position with a high collision profile might not be the best match. I would suggest trying, if possible, to match the playing position and the minutes of game time, as we know each playing position has a different physical demand. I'm not sure what benefit the orthopaedic controls will provide, unless the concussed player also sustained a musculoskeletal injury. They may provide a good comparison at the post-match time points, as they too will be removed from activity, but the effect of a moderate to severe musculoskeletal injury will overshadow these differences. In my opinion, the uninjured player would provide a better comparison to measure the effects of "sub-clinical" non-neurological injuries and musculoskeletal trauma. However, it would be interesting to see the differences in miRNA profiles between the injury types (concussion vs. musculoskeletal). The tenth paragraph of the study design section introduces a new aim that was not mentioned in the introduction. Specifically, to investigate the effects of collision sport involvement in baseline miRNA levels. I would suggest adding some rationale for this in the introduction. I would also consider the chance of false positives when comparing single extractions of miRNA. Because of the number of factors that influence miRNA expression, it is highly likely that the two baseline levels will be different. It would be bold to suggest that these differences are due to rugby participation alone. If one wanted to thoroughly investigate the baseline miRNA levels, one would ideally need to collect a number of samples to get an individual's "average range" that can then be compared. In the final paragraph of the study design section, the number of samples that will be sent to identify promising targets needs to be stated, even if this is a rough estimated number. How confident are the authors that this number of samples is sufficient to ensure that all the potential biomarkers are accurately identified? At the end of the of the outcome measures and statistical analysis section, it is mentioned that imaging data will be analysed. There is no previous mention of what imaging will be done or what it would be used for. Please clarify this.
--	--

VERSION 1 – AUTHOR RESPONSE

3) Pg 6, Ln 18: Complied? compiled?

Corrected

4) Page 8, Ln 6: ... is "a" form of

Corrected

5) I found it odd that all references are cited after a period or a comma, is it a requirement for this journal?

The authors noticed that in the recently published BMJ Open study protocols. Therefore, we decided to follow the same format.

6) Pg8, Ln 18-22: These two sentences need a little more clarification. For example, what are post concussion symptoms? migaine headache? cognitive functions such as learning and memory

problems? 80-90% of sufferers are able to return to pre-injury levels in terms of what? symptoms? cognitive functions? or physical activities?

Corrected

Abstract:

7) In the methods and analysis section, the three groups of players (uninjured, orthopaedic and concussed) could be described more clearly.

Corrected

Article summary

8) For the last point of the article summary, I would also include younger athletes as a population that requires further study, as this is an important group that displays differences in concussion symptom presentations. So I would suggest saying "such as female or youth athletes".

Corrected

Introduction

9) In the first sentence of the introduction, "sport related concussion" is capitalised, but in the abstract it is in lower caps and then abbreviated. This needs to be kept consistent.

Corrected

10) The first sentence reads: "Sport Related Concussion is form of mild traumatic brain injury caused by..", this should be changed to "Sport Related Concussion is a form of mild traumatic brain injury caused by..".

Corrected

11) In the second sentence there is a space between "England" and the punctuation.

Corrected

12) The third sentence needs to be grammatically revised. Possibly change to: "In the majority of cases, concussion results in a rapid onset, but short-lived, impairment of neurological function that resolves spontaneously."

Corrected

13) The fourth sentence is somewhat contradictory, as it presents two conflicting pieces of information. It is stated that 80-90% of concussed athletes return to pre-injury levels within two weeks, while 20-30% experience prolonged symptoms. Both statements can't be simultaneously true.

Corrected

14) The fourth sentence should also read: "pre-injury levels within two weeks"

Corrected

15) The second paragraph of the introduction is one long sentence. This could be split into two or three sentences.

Corrected

16) In paragraph five, "baseline information for athletes" can be changed to "baseline athlete information".

Corrected

17) In the first sentence of paragraph 6, the word "process" might not need to be capitalised.
Corrected

18) For the description of the HIA process in paragraph six, I would explain where it is currently being used after the first sentence and before going into a description of it. I feel that one needs the context of it being used in professional leagues to understand the description of video replays etc.
Corrected

19) In paragraph nine, I would add more biological information on why miRNA's are being suggested as biomarkers. For example, highlighting the temporal expression of miRNA and why/how quantifying these levels may provide an indication of the extent of underlying gene expression and the biological processes initiated by head injury. This may give the reader a better understanding of why miRNA analysis may be a better biomarker candidate in comparison to other gene or protein expression assays.

20) It might also be worth explaining how miRNA's are distributed in bodily fluids, how they interact with the blood brain barrier and the relevance of these dynamics to concussion.
In general, I think more detail needs to be provided about miRNA's and how they function to give readers without a knowledge of genetics or molecular biology a better understanding of why they are potential biomarkers.

19, 20) The authors think that would be beyond the scope of this study protocol and it would be better to refer the readers to our newly published paper "MicroRNA Signature of Traumatic Brain Injury: From the Biomarker Discovery to the Point-of-Care" for more details about the microRNA biology in TBI.

Di Pietro V, Yakoub KM, Scarpa U, et al. MicroRNA Signature of Traumatic Brain Injury: From the Biomarker Discovery to the Point-of-Care. *Frontiers in neurology*. 2018;9.

Methods and analysis

21) In the first sentence of the methods the number of participants is written as "1100", while in the sample size it is written as "1,100".
Corrected

22) In the first sentence, I would also add that the two leagues consist of senior professional players. Some international readers may not be familiar with the Premiership and Championship. For example, maybe say "the two highest tiers of senior professional domestic rugby competition", or something similar.
Corrected

23) Will the baseline miRNA samples be taken at a certain time of day to control for circadian fluctuations? Or possibly at multiple time points to provide an "average" baseline level? See Heegaard et al. 2016: "Diurnal Variations of Human Circulating Cell-Free Micro-RNA".

The authors are aware of the circadian fluctuations of miRNA. The investigators agreed to collect single preseason baseline sample before the training sessions, if possible, regardless of time of day. The authors had tried to collect the baseline samples at certain and fixed time of the day to control for circadian fluctuations but it was not practical for all the clubs due to time constraints and very busy training schedule preseason.

24) It will be important to record the timing of these baseline extractions in relation to preseason training sessions, other injuries, or supplement/medication use. There are so many potential confounding factors that getting a true "baseline" from a single extraction will always be difficult.

The investigators do record the timing of these baseline extractions in relation to preseason training sessions and other injuries. But, we do not collect information about supplement/medication use. The reason being that we have concerns we may not get reliable information from the players due to concerns about their confidentiality etc.

25) In the fourth paragraph, "in match HIA" could be changed to "pitch-side HIA" and "36-48 HIA" could be changed to "36h-48h HIA", to be consistent with Table 1.

Corrected

26) Paragraphs five and six could be joined into a single paragraph. Both paragraphs consist of a single, slightly long-winded, sentence.

Corrected

27) In the first sentence of paragraph seven, shown below, it is not especially clear which athletes are being referred to.

"In the case of players who are diagnosed with concussion at any time point, they will also be asked for a sample of saliva and urine at the point at which they are cleared by their team physician to return to rugby (Stage V) as per the Berlin Consensus Graduated Return-to-Play (GRTP) protocol.[2]"

The sentence could be made more succinct, for example: "Players diagnosed with concussion will be asked for a sample of saliva and urine at the point at which they are cleared by their team physician to return to rugby (Stage V) as per the Berlin Consensus Graduated Return-to-Play (GRTP) protocol.[2]"

Corrected

28) Will the playing position of the uninjured control be taken into account? For example, using a scrum-half as a comparison for a forward position with a high collision profile might not be the best match. I would suggest trying, if possible, to match the playing position and the minutes of game time, as we know each playing position has a different physical demand.

We have not asked clubs to match to playing position, just that they try and match a roughly similar number of played minutes in the game.

29) I'm not sure what benefit the orthopaedic controls will provide, unless the concussed player also sustained a musculoskeletal injury. They may provide a good comparison at the post-match time points, as they too will be removed from activity, but the effect of a moderate to severe musculoskeletal injury will overshadow these differences. In my opinion, the uninjured player would provide a better comparison to measure the effects of "sub-clinical" non-neurological injuries and musculoskeletal trauma. However, it would be interesting to see the differences in miRNA profiles between the injury types (concussion vs. musculoskeletal).

To rule out contamination from non-neurological injuries and musculoskeletal trauma (extra-cranial injuries) that would affect the sensitivity and specificity of miRNAs as reliable concussion biomarkers in polytrauma patients.

30) The tenth paragraph of the study design section introduces a new aim that was not mentioned in the introduction. Specifically, to investigate the effects of collision sport involvement in baseline miRNA levels. I would suggest adding some rationale for this in the introduction. I would also consider the chance of false positives when comparing single extractions of miRNA. Because of the number of factors that influence miRNA expression, it is highly likely that the two baseline levels will be different. It would be bold to suggest that these differences are due to rugby participation alone. If one wanted

to thoroughly investigate the baseline miRNA levels, one would ideally need to collect a number of samples to get an individual's "average range" that can then be compared.

Corrected

31) In the final paragraph of the study design section, the number of samples that will be sent to identify promising targets needs to be stated, even if this is a rough estimated number. How confident are the authors that this number of samples is sufficient to ensure that all the potential biomarkers are accurately identified?

Corrected

32) At the end of the of the outcome measures and statistical analysis section, it is mentioned that imaging data will be analysed. There is no previous mention of what imaging will be done or what it would be used for. Please clarify this.

Corrected

VERSION 2 – REVIEW

REVIEWER	Dr Sarah Mc Fie Department of Psychology University of Cape Town Cape Town South Africa
REVIEW RETURNED	01-Sep-2018

GENERAL COMMENTS	Overall, I believe the manuscript meets the requirements for the journal and details an important area of research. I noticed a few small things that could be addressed: In the first paragraph of the introduction, "short term" should be written as "short-term". When referring to Table 1, be consistent in using capital letters ("Table 1" vs "table 1"). The "University of Birmingham" abbreviation is described twice, once in the methods (paragraph 8) and once in the ethics and dissemination section. The abbreviation "SCRUM" should be defined in Table 1.
--